# High-Content *C. elegans* Screen Identifies Natural Compounds Impacting Mitochondria-Lipid Homeostasis and Promoting Healthspan

**DOI:** 10.3390/cells11010100

**Published:** 2021-12-29

**Authors:** Silvia Maglioni, Nayna Arsalan, Anna Hamacher, Shiwa Afshar, Alfonso Schiavi, Mathias Beller, Natascia Ventura

**Affiliations:** 1IUF-Leibniz Research Institute for Environmental Medicine, 40225 Duesseldorf, Germany; silvia.maglioni@IUF-duesseldorf.de (S.M.); naynaarsalan@gmail.com (N.A.); shafs101@uni-duesseldorf.de (S.A.); alfonso.schiavi@iuf-duesseldorf.de (A.S.); 2Institute for Mathematical Modeling of Biological Systems, Heinrich Heine University, 40225 Duesseldorf, Germany; anna.hamacher@hhu.de (A.H.); mathias.beller@hhu.de (M.B.); 3Systems Biology of Lipid Metabolism, Heinrich Heine University, 40225 Duesseldorf, Germany; 4Institute for Clinical Chemistry and Laboratory Diagnostic, Medical Faculty, Heinrich Heine University, 40225 Duesseldorf, Germany

**Keywords:** *C. elegans*, HCS, mitochondria, natural compounds, kahalalide F, lutein, neuroligin

## Abstract

The aging process is concurrently shaped by genetic and extrinsic factors. In this work, we screened a small library of natural compounds, many of marine origin, to identify novel possible anti-aging interventions in *Caenorhabditis elegans*, a powerful model organism for aging studies. To this aim, we exploited a high-content microscopy platform to search for interventions able to induce phenotypes associated with mild mitochondrial stress, which is known to promote animal’s health- and lifespan. Worms were initially exposed to three different concentrations of the drugs in liquid culture, in search of those affecting animal size and expression of mitochondrial stress response genes. This was followed by a validation step with nine compounds on solid media to refine compounds concentration, which led to the identification of four compounds (namely isobavachalcone, manzamine A, kahalalide F and lutein) consistently affecting development, fertility, size and lipid content of the nematodes. Treatment of *Drosophila* cells with the four hits confirmed their effects on mitochondria activity and lipid content. Out of these four, two were specifically chosen for analysis of age-related parameters, kahalalide F and lutein, which conferred increased resistance to heat and oxidative stress and extended animals’ healthspan. We also found that, out of different mitochondrial stress response genes, only the *C. elegans* ortholog of the synaptic regulatory proteins neuroligins, *nlg-1*, was consistently induced by the two compounds and mediated lutein healthspan effects.

## 1. Introduction

Aging is a complex and unavoidable biological process concurrently shaped by genetic, environmental and nutritional factors. The aging process is characterized by progressive accumulation of cellular damage, with consequent functional decline of different organs and systems, which leads to time-dependent increase in frailty and probability to develop pathologies and die. In the past 50 years, also thanks to the use of multicellular model organisms such as the round worm *Caenorhabditis elegans* (*C. elegans*) and the fruit fly *Drosophila Melanogaster*, several hallmarks of aging and many of their underlying genetic determinants have been identified. It is only in the past decade that research has been intensified to discover dietary interventions modulating the aging process.

Large-scale drugs and small molecules high-throughput- or high-content screenings (HTS/HCS) in search of enhancers or suppressors of selected readout(s) are conventionally carried out in single cell systems, i.e., culture cells, bacteria and yeast [1,2,3]. Screening of this kind have been instrumental to identify modulators of single hallmarks or determinants of aging [4,5,6,7]. However, since aging is associated with progressive deterioration of multiple tissues, organs, and physiological functions and cannot obviously be recapitulated at systemic level in single cells systems, the usefulness of multicellular organisms for screening in search of aging-modulators it is indisputable. *C. elegans* represents a powerful model which, due to its unique combination of features, such as its invariant cell lineage, its completely sequenced and highly conserved genome and its inexpensive and versatile laboratory handling, has been employed for whole-organism screening mainly to identify genetic factors affecting the aging process [8,9,10,11,12,13]. *C. elegans* aging can be easily manipulated not only with genetic but also with environmental, dietary and pharmacological interventions and a few screenings have been already carried out to identify extrinsic modulators of the aging process in *C. elegans* [11,13,14,15,16,17,18].

*C. elegans* lifespan, although much shorter than mice, still last an average of 20 days. Thus, rather than compound screenings throughout its whole lifespan, it would be ideal to use early surrogate markers of aging, which correlate with animal health/lifespan. In the past, some surrogate phenotypes have been proposed, such as animals increased resistance to stress or reduced adult body size, and they have been marginally exploited to identify pro-longevity interventions [11,13,19,20,21,22]. Here, we took advantage of our recently established *C. elegans* high-content microscopy platform [13] to screen a small library of natural compounds, including many of marine derivation, in search of pro-longevity interventions. After the initial automated screen in liquid culture, nine compounds have been identified which matched our screening criteria. Followed by validation through manual assays in solid agar plates, the four most relevant ones (i.e., lutein, kahalalide F, Manzamine A and Isobavachalcone) were selected for further analysis in *C. elegans* as well as in *Drosophila* cells. Of these four, we finally identified kahalalide F and lutein as two new compounds conferring increased resistance to stress and promoting health- and lifespan in a neuroligin-dependent manner.

## 2. Materials and Methods

### 2.1. Compounds

Compounds tested in our original screen were primarily extracted from marine organisms (e.g., sponges, algae, mollusks), and some were subsequently purchased for follow up studies. Specifically, lutein was purchased from Sigma Aldrich (St. Louis, MO, USA) (PHR1699, purity assessed by HPLC 88.8%) and dissolved in DMSO (10 mM stock solution). Kahalalide F was purchased from PharmaMar (Madrid, Spain) (411002, purity assessed by HPLC 92.3%) and dissolved in DMSO (5 mM stock solution). Manzamine A was purchased from Biomol (Cay21444-1, purity assessed by HPCE > 98%) and dissolved in DMSO (500µM stock solution). Isobavachalcone was purchased from Sigma Aldrich (SML1450, purity assessed by HPCE ≥98% HPCE) and dissolved in DMSO (10 mM stock solution).

### 2.2. C. elegans

#### 2.2.1. *C. elegans* Strains and Maintenance

We employed standard nematode culture conditions [23]. All strains were maintained at 20 °C on Nematode Growth Media (NGM) agar supplemented with *Escherichia coli* (OP50 or transformed HT115). Strains employed in this work were as follows: N2 (wild-type), CL2166 (dvIs19[pAF15(gst-4::GFP::NLS)] III), SJ4100 (zcIs13[hsp-6::GFP]), VC228 *nlg-1*(ok259) X and *cyp-14A4p::gfp::cyp-14A4 3′UTR* [24].

#### 2.2.2. Compounds Treatment

Worms were treated with the compounds in two different ways, either in liquid culture or on solid medium (NGM agar plates). All compounds were dissolved in DMSO, which was used as control in all the experiments. In liquid the maximum DMSO concentration used was 1% while in solid media was 0.25%. Different stock solutions with different concentrations were prepared for the compounds, thus allowing to have the same final DMSO concentration in control and compounds treated animals in liquid or solid media.

#### 2.2.3. Liquid Culture

In the first part of the screening *C. elegans* was cultured in liquid. The worms were synchronized by bleaching to obtain a large number of animals [25]. The screening was carried out on 96-well plates. Briefly, 10 mL of overnight grown OP50 bacteria were pelleted for 10 min at 4000 rpm; afterward, the supernatant was removed and the pellet was frozen in liquid nitrogen. Then, 10 mL of freshly prepared S-medium was prepared with the addition of following reagents: 8 μL Cholesterol (of 5 mg/mL stock solution in EtOH) and 50 μL 100× Penicillin/Streptomycin (P/S). The frozen bacterial-pellet was re-suspended with 4 mL of S-medium. The bacteria suspension was diluted in S-medium to obtain an OD600 = 2. We allowed the re-suspended bacteria to reach room temperature because lower temperatures could affect worms’ development. The final volume in every well of the 96-well plate was 250 µL, composed by: 140 µL of bacterial suspension, 10 µL compound under study plus H_2_O, 60 µL of S-medium, 40 µL of solution containing L1 larvae (1 worm/5 µL). Each column of the plate was used for a different concentration of the compound (border rows, A and H, and columns, 1 and 12, were not used). The plate was covered with a lid.

#### 2.2.4. Solid Medium

Nematode growth media (NGM) agar plates with HT115 bacteria were prepared and spotted with bacteria. Before the addition of compounds, bacteria on the plates were killed by UV exposure for 25 min. Afterward, the compound at the desired final concentrations was spotted. Egg-lay was performed on the same plates and the phenotype and GFP induction were checked on the third and fourth day after egg-lay. After spotting, plates were dried overnight at room temperature and used the day after.

#### 2.2.5. Multiwell Plates Preparation for Screening

Plates were prepared accordingly to our previously established protocol [13]. Briefly, on the day of the screen (3 days after plating and 4 after bleaching) plates containing animals treated with the compounds under study were washed with S-basal (duplicate plates per condition). Nematodes were transferred into 15 mL collecting tubes and allowed to settle by gravity. Two to three washing were carried out to remove excess bacteria which could interfere with the imaging. Worms were finally resuspended in S- basal and aliquots of 150 μL of solution were transferred in each well of a 96-well plate. Prior to image acquisition, worms were anesthetized by adding 2 μL of 1 M NaN_3_ to each well and the plate closed with a clear sealing film. Images were acquired with the ArrayScan VTI HCS Reader using a 2.5× objective and a 0.63× coupler using a 1-channel (GFP) assay. The Spot Detector BioApplication was used to quantify fluorescent intensity. Image acquisition was completed in about 25 min.

#### 2.2.6. Microscopy

A stereo microscope (modular, Leica MZ10 F Wetzlar Germany) with maximum 40× magnification was used to investigate the worm’s phenotype, specifically any delay in development or change is size and pigmentation was monitored. When using GFP reporter strains the pictures were taken using a color digital camera (0.5×) connected to the microscope, then GFP intensity, as well as the Red Fluorescence of Nile Red, was quantified using the software Image-J (http://imagej.nih.gov/ij/, last accessed on 7 September 2021).

#### 2.2.7. Quantitative Analysis of Nematode’s Body Size and GFP Induction of *gst-4*

For area and GFP intensity measurements the Cellomics ArrayScan Reader platform was used [13,26]. In this study, the BioApplication SpotDetector was used to quantify the effect of natural compounds on animals’ phenotypes, specifically the size and induction of *gst-4 or hsp-6*::GFP transgene.

#### 2.2.8. Fertility Assay

To analyze the effect of the compounds on worms’ fertility, the treatment was started from eggs. Gravid animals were allowed to lay eggs on plates containing the compounds for about 2 h, once ~40 eggs were reached the adults were removed and the eggs allowed to develop. As soon as the worms reached the L4 stage were individually transferred onto small (3 cm) NGM plates spotted with 50 μL of bacteria (HT115) at 20 °C. Animals were transferred to fresh plates every 24 h until day 7. Progeny in each plate was counted 48 h after hatching to facilitate the count of the larvae. The brood size of each animal was calculated as the sum of the total progeny in the whole fertile period.

#### 2.2.9. Heat Shock Assay

The treatment with the compounds started from eggs similarly to fertility and lifespan assay. On the first day of adulthood 20–25 animals were transferred into a fresh plate with the same condition and exposed to 35 °C in an air incubator. The plates were sealed with parafilm to avoid desiccation. Survival was checked every hour until the death of all animals (about 8 h). Survival rate was estimated by using Kaplan-Meyer estimator (OASIS 2 [27], available at http://sbi.postech.ac.kr/oasis, last accessed on 22 Novermber 2021).

#### 2.2.10. Toxicity Assay

Juglone sensitivity assay was carried out, with slight modifications, as previously described in [28]. Specifically, a stock solution of 50 mm juglone (Calbiochem) was freshly dissolved in DMSO prior to addition to molten NGM. Plates were freshly made approximately 3 h before use and seeded with concentrated OP50 30 min before being used. To assess survival, age matched young adult animals growth on HT115 NGM plates, were transferred to 60 mm NGM plates containing either compounds or control (DMSO) and assayed over time. Animals that escaped the plates were excluded from the analysis (censored). Animals were scored as dead if they did not respond to repeated light prodding. Percentages alive for each condition were determined by averaging the fraction alive per plate at each time point and plotting graphically. Survival curves were realized by using Kaplan-Meyer estimator OASIS 2.

#### 2.2.11. Lifespan Analysis

Survival curves and statistical analyses were carried out as previously described [29]. Briefly, survival analysis started from a synchronous eggs population and was carried out at 20 °C on solid NGM, at least 60 animals per condition were scored. Worms were treated with the compounds under study (or DMSO as control) from eggs throughout their lifetime.

Animals were scored as dead, alive or censored and transferred every day to fresh plates during the fertile period, and then every two or three days until death. Nematodes were considered dead when neither activity of their pharynx, nor response to touch was detected. Worms that died because of internal bagging, desiccation due to crawling on the wall of the plates, or gonad extrusion were considered censor. These animals have been included in the analyses until the time of censorship and are weighted by half in the statistical analysis. We used the Online Application for Survival analysis OASIS 2 for calculation of mean lifespan, standard deviation of the mean, and *p* values (Mantel–Cox regression analysis) from Kaplan–Meyer survival curves of pooled population of animals coming from at least two independent replicas. Data from survival analysis are summarized in Appendix A.

#### 2.2.12. Body Bend

Locomotion was assessed by counting the number of body bends per minute for each worm on solid agar plates with no bacteria. One bend was counted every time the mid-body reached a maximum bend in the opposite direction from the bend last counted. Body bends were checked in at least 10 single worms in 2 or 3 independent biological trials [30].

#### 2.2.13. Nile Red Staining

A stock solution (0.1 mg/mL) of Nile red was prepared dissolving the powder (Invitrogen N1142) in 100% acetone. The working solution was prepared in S-basal and added on top of NGM 6-cm plates seeded with OP50 bacteria, at a final concentration of 50 nM. The plates were previously spotted with the investigated compounds (or DMSO as control) and let dry. Synchronous eggs were allowed to develop and grow until the worms became adults on the Nile Red plates. Plates were always stored in the dark. Adult worms were transferred on plates without Nile Red 30 min before imaging. Then, 10–15 animals randomly selected from an initial sample of 40 worms were mounted on microscope slides. Lysosomes content was quantified by fluorescence microscope. Fluorescence intensity was measured through ImageJ software by selecting the same area between the vulva and the pharynx in each worm, on grayscale using the red channel of the acquired images. The assay was performed in triplicate.

### 2.3. Drosophila Cells

#### High-Content-Screening Experiments in Drosophila Cells

Drosophila S2R+ cells were cultured in Schneider’s medium with Glutamine in the presence of Penicillin/Streptomycin and 10% heat-inactivated fetal calf serum (all from PAN Biotech). For the compound tests, we followed previously established protocols [31]. In brief, cells were seeded in CellCarrier Ultra 96-well screening plates (PerkinElmer) in serum-reduced medium (5% FCS). Then, 10 mm compound stocks were serial diluted in DMSO to result in the final concentrations of 20 µM, 5 µM, 0.5 µM and 0.05 µM when diluted 1:500 (final DMSO concentration of 0.2% in all conditions) in the medium. DMSO only at the same concentration served as negative control. Cells were grown in the presence of 400 µM oleic acid (OA) to induce lipid storage over-night. For organelle detection, cells were stained with 10 µg/mL BODIPY FL C12, 150 nm TMRE and 5 µg/mL Hoechst33342 for 60 min and imaged using an Operetta CLS high content screening microscope (PerkinElmer) with 40× air objective magnification and in confocal mode. Three z-planes have been captured, each separated by a distance of 1 µm and processed as maximum intensity projections in further image analysis. On top of the fluorescence channels, we recorded digital phase contrast images (DPC) to report the general cell morphology.

Cell masks were generated by classification of the DPC images by the CellPose algorithm [32]. Subsequently, masks were imported in the CellProfiler image analysis software [33] and cell specific parameters were determined by segmentation of the individual channels using custom analysis pipelines (available from the Lab’s GitLab account at: https://gitlab.com/Beller-Lab, last accessed on 25 Novermber 2021). Numeric data was preprocessed with KNIME ([34] KNIME pipelines available from: https://hub.knime.com/matbeller/spaces/, last accessed on 25 Novermber 2021) and plotted with GraphPadPrism.

### 2.4. Statistical Analysis

Data are represented as mean ± SEM from at least three independent biological replicas carried out in a blinded manner where possible. Statistical analyses were performed using either two-sided Student’s *t* test or one-way ANOVA as specified in each figure legend. GraphPad Prism 8 software was used for all statistical analysis to calculate significance as follows: * *p* = 0.01 to 0.05, ** *p* = 0.001 to 0.01, *** *p* = 0.0001 to 0.001, and **** *p* < 0.0001 versus respective control conditions. Survival experiments were analyzed using OASIS2.

## 3. Results

### 3.1. A Phenotype-Based Automated Screen Identified Four Natural Compounds with Potential Anti-Aging Effects

In the nematode *C. elegans*, it is widely established that different degrees of mitochondrial stress (mitochondrial hormesis) achieved through genetic, silencing or pharmacological suppression of the mitochondria electron transport chain (ETC), lead to opposite healthspan effects concurrently associated with very reproducible phenotypic features (the so-called *Mit* mutants’ phenotype). Namely, “mild” ETC suppression extends lifespan, prolongs animal developmental timing and fertility period, reduces adult size and increase paleness; “severe” ETC disruption instead induces animals’ sterility, growth arrest or lethality [10,19,35]. Moreover, regardless of the degree of ETC disruption, this affect the expression of mitochondrial stress response (MSR) genes, such as antioxidants (e.g., *gst-4*) and the mitochondrial unfolded protein response (e.g., *hsp-6*) [35]. Taking advantage of these discrete and highly reproducible phenotypic features, we established an automatic phenotype-based microscopy pipeline to identify substances with potential beneficial (pro-longevity) or detrimental (toxic) effects acting through mitochondria or MSR pathways [13]. In this study, we exploited this platform to screen a sub-library of 25 extracts of natural origin, many of marine derivation (Appendix A), searching for those inducing the stereotyped *Mit* mutants’ phenotype, and thus with potential anti-aging activity (Graphical abstract). Among the compounds under study some have known antioxidant (e.g., homosekikaic acid and lutein), anti-inflammatory (e.g., helenalin and isovitexin) or anti-cancer (e.g., kahalalide F or kuanoniamine D) activity.

Specifically, we used the Cellomics ArrayScan^®^ VTI HCS Reader imaging platform, with a liquid handling setting, to first identify compounds that reduced animal size and increase the expression of MSR genes (*gst-4* or *hsp-6*) [29,35] at low concentrations, while being lethal or arresting animal development at higher doses. A preliminary compound testing (only using the *gst-4* reporter strain) had excluded nanomolar, non-effective concentrations of compounds and we then carried out the primary screening in liquid culture using a micromolar range (1, 10 and 100 µM) in the two different transcriptional fluorescent reporter strains. With this liquid setting, nine compounds were identified for their ability to affect nematodes’ size and GFP intensity to different degree in the two reporter strains (Figure 1 and Figure 2). All other treatments did not result in any significant change (Appendix A).

Compared to control treatment (1% DMSO), animals’ body size was significantly reduced by 100 µM homosekikiac acid, 1 µM kaemperol-3-o-rutinoside, 1 µM kaempferitin, 10 µM Kuanoniamine D, 100 µM lutein, 100 µM lupeol and 100 µM macrosporin (Figure 1a,e,f,g,h and Appendix A); while 100 µM isobavachalcone, 1, 10 and 100 µM kahalalalide F and 1, 10 and 100 µM manzamine A, 10 and 100 µM 8-OH-manzamine A, resulted in early L1-developmental arrest, and the small larvae were not detected by the system (Figure 1b,d,i and Appendix A). When checking the expression of the two mitochondrial stress fluorescent reporters, *gst-4* and *hsp-6*, due to compromises in optimization of screening detection parameters, when nematodes were too small to be detected by the BioApplication, no GFP intensity could be measured (Appendix A and Figure 2b,d,i). Regardless of these difficulties, quantified data showed that some of the treatment lowered the expression of *gst-4*, namely 1 and 10 µM hexaprenylhydroqinone, 100 µM homosekikaic acid, 10 µM 2-hydroxy-4-methoxyphenylacetonitrile, 1 µM isovitexin, 1 µM kaempferol-3-rutinoside, 1 and 10 µM 8-OH-manzamine A (Figure 2a,c,e and Appendix A). Other treatments did not show significant results, although some displayed a trend in *gst-4* expression (Figure 2b,f,g,h and Appendix A). Conversely, 10 µM lupeol significantly increased the induction of the *hsp-6* (Appendix A). The screening platform utilized in this study to search for compounds inducing mitochondrial hormesis, was previously developed and validated with genetic and pharmacological interventions specifically targeting different subunits of the ETC complexes (such as oligomycin) [13]. Here, using oligomycin as a positive control, we revealed inconsistent effects on animal size (Appendix A), likely ascribed to differences in the genetic background of the two mitochondrial stress reporter strains (*gst-4* and *hsp-6*) or to mild size changes which fall below the threshold of detection of the automatic microscopy platform. Instead, clear alterations of transgenes expression were observed (Appendix A), which consistent with previous findings do not linearly correlate with the severity of the mitochondrial stress treatment [35].

To confirm the Cellomic results in size and gene expression we thus also visualized the 96-well plates under a fluorescence stereomicroscope. Cross-comparison analysis of size and GFP intensity between unbiased (Cellomic) and operator sensitive (stereomicroscope) methodologies led us to select nine compounds (Figure 1 and Figure 2) for validation in a second step with refined compounds concentrations. The validation step was carried out in solid nematode growth medium (NGM), the experimental set up in which age-associated phenotypes would be eventually characterized. For three of the compounds leading in the liquid screening to growth arrest or lethality, we reduced the final concentrations to be tested in the next step to see whether we could simply reduce animals’ size or slow down development (*Mit* phenotype). Specifically, we used 25 and 50 µM isobavachalcone, while 0.1 and 0.5 µM kahalalalide F and manzamine A. Wild-type nematodes fed HT115 bacteria in NGM at 20 °C began laying eggs on the 3rd day after hatching. This validation phase clearly confirmed that some compounds affect nematode’s development, which was slightly slower compared to untreated animals, and the F1 progeny only appeared on the 4th or 6th day of treatment (Table 1): 100 μM homosekikiac acid, both concentrations of isobavachalcone and kuanoniamine D, as well as 0.5 μM manzamine A. Other compounds instead only had minor but still relevant effects: 0.5 μM Kahalalide did not altered development but reduced animal size compared to control condition; lutein at the highest concentration only slightly reduced animal’s pigmentation (Table 1). Based on the observed phenotypic differences (in the first/liquid media and second/solid media assays) induced by the nine compounds on animals’ size and transgenes expression, as well as on their availability, four compounds (Figure 1) were chosen for further characterization of their effects on worms’ healthspan associated parameters: isobavachalcone, kahalalide F, manzamine A and lutein.

### 3.2. Validation of Four Compounds on Cellular and Animals’ Physiology

The effect of the four compounds was then first investigated on worms’ fertility, a health-related parameter of the germ cells often correlating with mitochondrial-driven changes in lifespan [19]. Worms were supplemented from embryos with different concentrations of isobavachalcone (10 or 25 µM), kahalalide F (0.5 and 1 µM), manzamine A (0.1 and 0.5 µM), lutein (10 and 100 µM) or 0.25% DMSO as control and the number of laid embryos per worm per hour as well as the total brood size were calculated across the entire fertile period. At the highest concentrations used, isobavachalcone and manzamine A reduced the fertility rate below 45% of control (Figure 3a,b,d,e), while a moderate reduction of fertility (although not statistically significant) was achieved with the highest doses of kahalalide F and lutein (Figure 3g,h,j,k). Mild mitochondrial stress extends lifespan and concurrently reduces the amount of fat content [36]. Thus, we then tested whether animals’ lean phenotype observed upon compound treatment correlated with the amount of Nile red, which stains *C. elegans* lysosome-related organelles, and nicely correlates with Oil red staining of fat compartments upon MMS [36]. To this end, we grew animals on plates containing the four hit compounds and Nile red [37] and found that all treatments led to a significant reduction of Nile Red signal, especially the highest dose of Manzamine (Figure 3c,f,i,l and Appendix A).

A wider number of marine-derived compound was independently screened in Drosophila S2R+ cells through a high-content microscopy assay, specifically looking for their impact on lipid content and mitochondria activity (Beller, *unpublished data*). Very interestingly, the four compounds identified in the *C. elegans* screen were also within the hits of the cells screen. The four lead compounds were therefore tested in Drosophila cells at different concentrations (ranging from 20 to 0.05 μM) revealing differential effects on mitochondria and lipid parameters (Figure 4a and Appendix A). Mitochondrial membrane potential was assessed with TMRE and lipid droplet (LD) deposition, induced by feeding with OA (400 µM), was detected by counterstaining with Bodipy FL C12. We found that besides lutein all compounds resulted in a reduction of cellular lipid storage levels with highest potency for Manzamine A (Figure 4b). Isobavachalcone, kahalalide F and manzamine A also showed the strongest impact on mitochondrial membrane potential (Figure 4c). The latter, however, also affected cell proliferation at high concentrations (Figure 4d), inducing overall cell and nucleus altered morphology (Figure 4a, DPC), which is in line with previous reports [38]. Based on the combined in vivo and in vitro results, we only selected for further studies the two compounds, kahalalide F and lutein, which less severely affected cellular and animals’ physiology.

### 3.3. Kahalalide F and Lutein Promote Animals’ Healthspan

We investigated the ability of the two compounds to specifically increase resistance to heat or oxidative stress, which, in *C. elegans*, are often associated with lifespan extension [20]. When supplemented from embryos and through the entire life, both compounds protected adult nematodes against heat stress-induced lethality. Namely, control DMSO fed animals reached 50% of mortality after 6.5 h at 35 °C. At this time point animals treated with kahalalide F and lutein were still more than 75% alive and they reached 50% mortality only after 7.5h and 7h respectively (Figure 5a,b). Similarly, and to a bigger extent, the two compounds significantly increased animals’ resistance to juglone-induced oxidative stress. The mortality rate reached 50% in the control animals already after 1.15 h on juglone plates. At this time point, animals treated with kahalalide F and lutein were still more than 75% alive and they reached 50% only after 3.37 h and 2.7 h respectively (Figure 5c,d).

The two compounds were then tested for their ability to extend lifespan. We found that both lutein and kahalalide F significantly extend *C. elegans* lifespan (15% and 10% increase compared to control respectively, Figure 5e; Appendix A) and healthspan, assessed as the ability to move throughout the entire life (Figure 5f and Appendix A).

### 3.4. Lutein Pro-Health Effects Are Mediated by nlg-1 Induction

To start investigating molecular mechanisms underlying the beneficial effects of the two compounds, we looked at the expression of genes involved in heat and oxidative stress responses. The FOXO/DAF-16 transcription factor translocates to the nucleus to mediate the increased resistance to stress upon different pro-longevity interventions [39]. Lutein and kahalalide F, however, did not affect DAF-16::GFP nuclear translocation neither in basal conditions nor upon heat-stress at any of the observed time points (Appendix A). Based on the induction of the classical *Mit* phenotype the two compounds may promote healthspan by triggering MSR, which are known to work independently of *daf-16* [40,41]. We therefore assessed the expression of classical MSR genes, such as *gst-4, hsp-6* [29], which to some extent, were affected in the high-content screening. To our surprise their expression was not significantly increased by lutein or kahalalide F (Figure 6a,b) although these may reflect a dosage specific effect and the fact that their induction is not necessarily associated with lifespan extension [35,42]. We also checked the expression of *cyp-14A4,* a detoxification gene induced as part of cytoprotective response induced upon mitochondrial stress [24], which was also not affected, at least by the used concentration of lutein and kahalide F (Figure 6c). Mitochondrial stress extends lifespan through neuronal signals [23,43]. Neuroligins (NLG1-4) are a family of neuronal genes involved in the regulation of synaptic functionality [44]. In *C. elegans*, there exists only one neuroligin homolog, *nlg-1*, which is induced by the redox transcription factor Nrf2/SKN-1 to mediate resistance to oxidative stress [28] and that we recently found to be induced by mitochondrial stress [45]. Interestingly, lutein and kahalalide F significantly increased the expression of *nlg-1* (Figure 6d and Appendix A), and, most notably, *nlg-1* knockout completely prevented the beneficial effects on lifespan and movement induced by lutein (Figure 6e,f).

## 4. Discussion

In summary, using our in vivo phenotypic-based screening pipeline, we were able to identify two new natural compounds of marine origin, kahalalide F and lutein, promoting health and lifespan in *C. elegans* (Graphical Abstract)*,* the latter in a *nlg-1*-dependent manner. Our screening method in liquid presents with some caveats. For the automatic detection and measurement of the GFP intensity we had to compromise settings optimization between the low magnification and the exposure time to recognize the highest number of animals per well. As a drawback, nematodes that were too small (arrested as L1) could not be detected by the BioApplication, and therefore no GFP intensity could be measured. These caveats were nonetheless bypassed by coupling the initial screening with observation of the screening plates under epifluorescence microscopy. Moreover, according to the notion that exposure in liquid vs. solid medium can impact compounds bioavailability and effects, we also observed that exposure of animals in liquid to oligomycin, our positive control, did not consistently increased the expression of the two GFP reporters as it did in agar plates [13]. To overcome this problem, we validated the nine more promising compounds from the initial in liquid screening step, in NGM agar plates. This step led to the selection of four promising compounds which were further tested on animal’s health related parameters such as fertility and fat metabolism. Along with the availability of the compounds (manzamine A and isobavachalcone would require additional testing to refine their optimal beneficial concentration), this led to the identification of two compounds, kahalalide F and lutein as novel promising natural compounds with anti-aging effects, possibly working through mitochondrial hormesis, thus ultimately supporting the usefulness of our screening platform to identify pro-longevity interventions. Our screening strategy was further validated by assessing the effects of the four hits in fly cells in mitochondria-lipid homeostasis and cell viability.

The protective MSR pathways must be activated early during development in the *C. elegans Mit* mutants to extend lifespan [19,35,46]. We hypothesize that the same protective pathways, when identified, may be activated to prevent or delay the devastating effects induced by severe mitochondrial deficiency [47]. In support of our hypothesis, the compounds identified here to induce the typical *Mit* phenotypes—and thus likely able to extend lifespan through induction of MSR pathways—were also identified in a different phenotype-based screening (but at different concentrations) in search of possible therapeutics for mitochondrial complex I deficiency associated diseases [45]. The validity of our screening platform is therefore supported by the fact that two different types of screens, looking for compounds possibly activating MSR pathways, identified the same compounds. Here, we specifically found that lutein and kahalalide F increase animals’ resistance to heat and oxidative stress and promote healthy aging.

Lutein is a carotenoid belonging to the xanthophyll family, which can be found in leafy dark green plants such as spinach, kale and marigold as well as in certain microalgal species such as *Scenedesmus almeriensis*, *Chlorella zofingiensis*, and *Muriellopsis* sp. [48,49]. Moreover, lutein is often referred to as the “eye vitamin” because of its localization and protective role against light-induced damage in the human eye retina [50]. The most probable natural function of this pigment in macular membranes is the protection against oxidation and radiation-induced damage of lipids [51,52,53]. The beneficial effects of lutein seem however to go well beyond its protective role in the eye, in fact, a number of studies points out its positive effects in several human pathologies, ranging from cancer to neurodegenerative disorders and these may not always rely on its antioxidant properties [54,55,56,57]. Evidence shows, for instance, that lutein can block the activation of NF-kB, a key player in the generation of immune response [58], and the degradation of the inhibitor kB (I-kB) [59]. When lutein dissociates the I-kB from NF-kB, this can translocate into the nucleus, where decreases gene transcription and synthesis of inflammatory markers such as cytokines, chemokines, and iNOS [60]. Kahalalide F is instead a marine-derived cyclic- peptides, extracted from mollusks and algae, with promising applications in cancer therapy, although its mode of action is largely unknown [61,62,63,64]. Kahalalide F was shown, for instance, to induce cell death via oncosis preferentially in tumor cells and that this effect, in line with our findings, is achieved through a loss of mitochondrial membrane potential and lysosomal integrity [65,66]. In search of possible MSR activated by the two identified compounds, we found that *nlg-1* is induced by both compounds and mediates the beneficial effects of lutein. This finding hints to a beneficial effect of lutein (and kahalalide F) through the activation of specific neuronal MSR pathways, or alternatively implies expression or effects of NLG-1 in tissues other than neuronal synapses. Of note, both mitochondria and lipid homeostasis play a crucial role in synaptic functionality [67,68].

In further support of our above-mentioned hypothesis, in a different study [45], we identified *nlg-1* as a target of lutein in a *C. elegans* model for mitochondrial complex I deficiency associated disease. However, in this disease model, lutein was used at a much lower concentration and provided its beneficial effects by reducing the increased expression of *nlg-1* associated with severe mitochondrial disruption. Here instead, we found that the pro-longevity effect of higher doses of lutein is mediated by increased expression of *nlg-1*, likely through a mild mitochondrial stress [69]. Clearly, lutein may act through different mechanisms in a dose and context dependent manner and it is involved in fine tuning the expression of this synaptic regulatory gene, which may play opposite roles in response to mild or severe mitochondrial stress. Although the precise mode of action of the identified compounds will require further investigation, our work indicates that neuronal synaptic functionality is implicated in mediating healthspan outcomes in response to different degrees of mitochondrial stress. In conclusion, our findings clearly suggest that natural compounds act through mitochondria hormesis to promote healthspan. Moreover, they bring further support to the notion that neurons are critical mediators of mitochondrial stress control of longevity. It will be interesting to explore whether the additional compounds identified in our screen can also extend lifespan and/or protect against mitochondriopathies and to further investigate their underlying molecular mechanisms.

## Data Availability

For all results presented in this work, a “DataSource” Excel file is available online. For Drosophila Cells custom analysis pipelines are available from the Lab’s GitLab account at: https://gitlab.com/Beller-Lab (last accessed on 25 December 2021). KNIME [34] pipelines are available from: https://hub.knime.com/matbeller/spaces/ (last accessed on 25 December 2021).

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
