# Peer review of "High-Content C. elegans Screen Identifies Natural Compounds Impacting Mitochondria-Lipid Homeostasis and Promoting Healthspan"

_cells, 2021, doi:10.3390/cells11010100_

Round 1

Reviewer 1 Report

Maglioni et al. have screened natural compounds to discover novel genes contributing to C. elegans longevity. In a liquid culture screen, the authors first sought to identify those agents that affect worm size and expression of mitochondrial antioxidant (gst-4) and mitochondrial unfolded protein response (hsp-6) genes. In a second screen on solid agar plates, they narrowed down the list to four candidate compounds that displayed the most profound effects on reducing C. elegans development, fertility, and its capacity to store lipids. Similar observations were derived in Drosophila cells S2R+, suggesting that these substances could activate stress response pathways in another species. Finally, the authors proceeded with the final two hits, kahalalide F and lutein, to demonstrate that these drugs play essential roles in conferring augmented thermal and oxidative resistance, and promoting a longer lifespan (minor increase) and healthspan in aging C. elegans. Furthermore, they revealed that these phenomena occurred in a neuroligin-dependent manner following lutein treatment, suggesting the activation of previously unknown mitochondrial stress response pathways in neuronal populations.

Overall, the data support the proposed conclusions.  There are numerous typos and mistakes. The figures can be improved to ease the understanding of the data (e.g. data in table 1 is poorly designed). The body of experimental evidence could be strengthened further if the authors will have provided a proof that their initial screen is titratable with respect to the final four natural compound library targets.

Major point:

Although it is understandable that the authors have performed primary screening in C. elegans with fixed concentrations of the drugs tested, the half-maximum doses of the final four hits were not evaluated in terms of affecting worm size and GFP expression. Please provide at least simple drug titration curves for isobavachalcone, manzamine A, kahalalide F, and lutein demonstrating that their effects on both worm size and GFP fluorescence can be more quantitatively titrated in contrast to rather qualitative data presented in Figures 2,3,S1,S2.

It is not clear to me why Drosophila cells are used instead of mammalian cell lines to test the effect of compounds. Please explain.

Minor points:

There are too many typos and mislabels in figures and figure legends. I decided not to list any of them here. This manuscript needs to be professionally edited. Below I list items that need further explanations or better description.

  • It is not clear why the authors have restricted their screen to only compounds beginning with "H", "I", "K", "L", and "M" (Table SI)?
  • Please include a subsection on how statistical analyses were conducted as part of the Materials and Methods section.
  • "These rows can be used to indicate the strains and the condition of treatment" (line 116) is puzzling since it refers to rows while the preceding sentence "The border columns were not used to avoid the contamination" refers to columns. Please fix.
  • Please specify the type of "control" in "To assess longevity, age matched young adult animals were transferred to 60 mm NGM plates containing either drug or control and assayed over time" (line 178).
  • Please explain in the text why oligomycin had a positive effect on GFP intensity in the hsp-6::GFP strain in Figure S2b but not in Figure S1b?
  • It is not clear what is the difference between the datasets presented in Figure 2 and Figure S1 and Figure 3 and Figure S2? Please explain in the text.
  • From "While some treatments didn‚Äôt result in any significant change in the first round of screening in liquid, other compounds were able to affect nematodes‚Äô size and GFP intensity to different degree" (line 274) is not clear to what screening the authors refer as "the first round of screening"? Was this the "preliminary testing" or "the screening with higher doses‚Äô range (1, 10 and 100 ŒºM)" mentioned in "A preliminary testing (using gst-4 reporter strain) excluded lowest, non-effective concentrations and we continued the screening with higher doses‚Äô range (1, 10 and 100 ŒºM) using the two different transcriptional fluorescent reporters (Figure S1a,b; Figure S2a,b)"?
  • From the color-coded differences indicated in Table I is not clear whether they are statistically significant?
  • From the legend to Figures 4, S1, S2, S3, S4 and Table SII is not clear whether worms were grown on solid or liquid medium during to the respective experiments?
  • Please enlarge and increase pixel density of panels presented in Figure 5a so that individual morphological features and cellular organelles are clearly visible.
  • Could the authors please add a new concluding figure with a schematical description of the effects of kahalalide F and lutein modulation on aging in C. elegans? This can also involve illustration of the Mit mutants‚Äô phenotype.
  • "Results of the automatic screening carried out with the Cellomic ArrayScan reader" (line 524) seems to be identical between Figure S1 and Figure S2 in the Supplementary Material description section. Please provide a more refined description for each supplementary figure.
  • It is not clear whether error bars are so small or were not plotted for Isobavachalcone (100 ŒºM), Manzamine A, 8-OH-manzamine A (10, 100 ŒºM) in Figure S1a; Kahalalide F, Lufferiellolide (100 ŒºM) in Figure S1b; Isobavachalcone (100 ŒºM), Manzamine A, Manzamine F (10, 100 ŒºM) in Figure S2a; Kahalalide F, Kojic acid (100 ŒºM), Lufferiellolide (100 ŒºM) in Figure S2b?
  • It is also not clear what does asterisk indicate in "8-OH-manzamine A*" Figures S1a, S2a?
  • From the legend to Figures S1, S2 is also not clear what was the purpose of testing oligomycin? In addition, the indication of whether measurements represent mean ¬± SEM, what statistical test was used for comparison, what do asterisks "*" and "ns" denote, and p-value definitions is also missing.
  • It is not clear what does the pink box indicate in Figure S3? Please mention in the respective figure legend.
  • Please provide statistics for Figure S6a to demonstrate that the observed differences were indeed significant.

Reviewer 2 Report

Summary

The manuscript of Maglioni et al. presents a study of the effect of various compounds on C. elegans to assess life span and healthspan. The article starts with 25 compounds that are reduced to a principal set of 9 studied in depth. Beginning with the markers gst-4 and hsp-6, the authors used lipid droplet deposition, mitochondrial membrane potential and cell count in a Drosophila model to narrow the compounds to two compounds of interest. The two compounds, Lutein and Kahalalide, were assessed for lifespan and healthspan effects in C.elegans, significantly increased in both treatments with the compounds via an nlg-1 dependent mechanism.

Brief Comments

The article is well written, clear in its focus and presented logically. There are some specific areas where the authors could improve the manuscript, but for the most part, the work is sound. The only general concern is the rationale for switching to a Drosophila model after initial assessment in C.elegans. This switch in the model organism is not particularly well detailed, and thinking in more broad terms could be better transitioned in the text for the reader.

Specific Comments

Some areas can specifically be improved by the authors:

  • 3 The formatting of the procedure is not in keeping with the formatting of the research article and the authors should seek to bring consistency to this point.
  • Line 160 the phrase ‘a sufficient number of cells’ is too vague and there should be an indication of time or at the very least an approximate number of cells.
  • Line 232 the authors should indicate if there was any washing after the staining or removal of the animals from the staining solution. This is not clear and should be clarified in the methods.
  • At line 312, the authors highlight the use of the stereomicroscope, what was not clear as a reader, why was this data not shown for comparison? This point here should be clarified in the results section.
  • At line 315, this sentence really ought to appear earlier in the manuscript. At this point, the reader has only exclusively seen data for these nine compounds in Figures 2 & 3. Therefore it would be logical for this to appear earlier in the text.
  • Line 338 the selection criteria for the 4 compounds should be clarified for the reader.

Minor points

  • Line 46- For High Content Screen and High Throughput Screens the abbreviation should be made clear in the text, perhaps at line 44
  • Line 125- NGM has already been introduced and the full written out text should appear at line 97 with NGM for the first time.
  • The reference for OASIS at line 201 should be placed at line 172.
  • The use of a comma instead of the decimal point should be corrected, for instance at lines 279, 285, 396, 398, 402, 404

Figures & tables

  • Figure 1- there is no need for the letters a-d to label the panels, the letters are not referred to in the legend and each of the compounds is clearly named.
  • Table 1- is not effective, the figure should be simplified or perhaps the authors should substitute it for a heatmap. Generally, the table and its construction needs to be reviewed.
  • Figure 4- the presence of the names of the compounds above each panel is not needed as in each row they are the same. Perhaps the authors should consider placing the name across all three panels in each row.
  • Figure 5a- why are the images for the remaining compounds not shown? I think this is important, if the images are to be shown in the main text then the other compounds should be as well.
  • Figure 5c- the title for this should be TMRE fluorescence and not mitochondrial, as rightfully TMRE is being measured or also possible is mitochondrial membrane potential.
  • Figure 5 legend- DPC should be written out fully to ensure that the legend is in fact independent of the text.
  • Figure 6a&c, the inclusion of juglone and temperature on the x-axis is not needed and could be removed to make all panels uniform and place these details in the title of the panel .

Reviewer 3 Report

Maglioni et al reports a screening of natural compounds to identify possible antiaging products. To do that, the authors use a high-content microscopy platform and look for phenotypes associated to mild mitochondrial stress, which is associate to animal’s health and life span. First, they use the platform to use different concentrations of 25 compounts, and later the authors validate the 9 more promising drugs in solid medium. This step led to select four compounds that were tested on animal’s health related parameters. Finally the authors selected two of them, kahalalide F and lutein as novel promising natural compounds with anti-aging effects, possibly working through mitochondrial stress response pathway and probably mediated by nlg-1 (neuronal gene involved in the regulation of synaptic functionality and involved in resistance to oxidative stress).

The novelty of the paper is good and the experiments are well conducted, however, some questions should be clarifique and some mistakes should be corrected.  The main concerning is the control used in different experiments: ranking DMSO 0,25%-1%. I think the appropriated control should be the amount used in each of the natural compounds, because DMSO probably affect the phenotypes.

Results 1) A bit messy. Could be great if is possible reorganize a bit better and following the order from fig and S.

I think there are some mistake:

Line 2. C. Elegans should be C. elegans

Line 24. Drosophila should be Drosophila

Line 41. Drosophila melanogaster

Line 46. HTC/HCS write meaning

Line 52. I think model could be better that platform

Line 70. Drosophila

Line 89. C. elegans

Line 91. Escherichia coli

Line 162. Why HTT115 is used instead of E. Coli OP50?

Line 191. Control 0.25% DMSO, why? The natural compounts are solved in different DMSO concentration. Should be the same amount of DMSO in control and treated.

Line 195. Why animals with internal bagging, or gonad extrusion were censored.

Figure S2. Normalized 1%DMSO?

Line 276. In Fig 2 and Supplementary, there are differences in some of the compounds and sometimes appear as significatives or not. Example: Homosekikiac acid (Supplementary ns and in Fig2 as significative at 100 microM), Isobavachalcone 1 microM ,Kuanoniamin 10 microM (Fig 2 sign and Suppl ns), Lutein 10 microM, (Fig 2, ns, Supp sign), Macrosporin 10 microM (Fig 2, ns, S sign).

Same happen with GFP intensity, there are differences between results obtained in Fig 3 and S2. Homosekikaic acid, isobavachalcone, isovetexin, kuanoniamin D

Line 292. 10 microM lupeol. Is it 100??

Line 307. g and h?

Line 320. Feed   with HT115, why? Why not OP50?

Results 3.1. In general looks a bit messy

Line 342. Kuanoniamin D???

Line 344. Same order you show in picture (Iso and manzamine)

Line 356. Title I think should be include Nile Red Staining too.

4 or four

Fig 6c. At 4 and 5 hours error bar are missing

Line 430/439. Knockout or Knock-out

Line 557/582. References are duplicated (8 and 22)

Line 565/584. References are duplicated (12, 23)

Fig S6. Lutein/lutein

Fig S7. Fig8d?? In S7 is said quantification is shown in Fig 8d, but Fig 8d doesn´t appear

Round 2

Reviewer 1 Report

Maglioni et al. have considerably improved their manuscript and successfully explained all contentious issues. There are only a few minor points regarding the graphical abstract to be dealt with.

Minor points:

1) The new Graphical Summary of the results is a welcome asset, however, the icons used are too small. Please enlarge particularly all small compound, worm, fly, and cell icons. It might be also a good idea to slightly increase the size of the "Size" and "GFP" graphs as well as that of the Falcon tubes. In addition, the 12-well plate is relatively smaller than the 96-well plate. Please fix.

2) The grey dashed lines outlining the Graphical Summary of the results are not aesthetic. Please use continuous border line formatting. Also, please do not draw these border lines overlapping with the grey headline arrowheads and horizontally align them so they become exactly juxtaposed under the grey headline arrowheads.

3) Please remove bold formatting from ") Hits identification.", ") Hits validation.", and ") Hits selection." in the legend to the Graphical Summary of the results.

4) Please change "Heathspan" to "Healthspan" in the legend to Figure S7.
